# Differential contrast enhancement using conditional deep learning for Gadolinium dose reduction in brain MRI

**Thomas Pinetz**[*1]                                                PINETZ@IAM.UNI-BONN.DE
[1] *Institute of Applied Mathematics, University of Bonn, Germany*

**Erich Kobler**[*2]                                                ERICH.KOBLER@UKBONN.DE
**Robert Haase**[2]                                                ROBERT.HAASE@UKBONN.DE
[2] *Department of Neuroradiology, University Hospital Bonn, Germany*

**Julian A. Luetkens**[3]                                                JULIAN.LUETKENS@UKBONN.DE
[3] *Department of Diagnostic and Interventional Radiology, University Hospital Bonn, Germany*

**Alexander Radbruch**[2,5]                                                ALEXANDER.RADBRUCH@UKBONN.DE
[5] *Clinical Neuroimaging, German Center for Neurodegenerative Diseases (DZNE), Bonn, Germany*

**Katerina Deike**[2]                                                KATERINA.DEIKE-HOFMANN@UKBONN.DE
**Alexander Effland**[1]                                                EFFLAND@IAM.UNI-BONN.DE

## Abstract

In this work, we propose a novel deep learning (DL)-based method to reduce the dose of Gadolinium-based contrast agents administered in brain MRI examinations. In contrast to recent DL approaches, we explicitly focus on accurately predicting contrast enhancement signals and synthesizing realistic images, leveraging contrast signals from subtraction images of pre- and post-contrast T1-weighted image pairs. By training our model to only extract and enhance contrast signals, and by conditioning its layers on relevant physical parameters, we demonstrate its effectiveness across diverse datasets, including data at different dose levels from various scanners, field strengths, and contrast agents.

**Keywords:** Deep Learning, MRI, GBCA reduction, Neuroradiology.

## 1. Introduction

Gadolinium-based contrast agents (GBCAs) play a vital role in diagnosing neuropathologies, yet they pose challenges due to cost, patient discomfort, potential health risks, including nephrogenic systemic fibrosis (Kanda et al., 2014; Schieda et al., 2018), and environmental pollution (Brünjes and Hofmann, 2020). Current guidelines advocate the reduction of GBCA dosage while still enabling reliable diagnosis (ACR Committee on Drugs and Contrast Media, 2023). To address this challenge, recent DL-based methods (Pasumarthi et al., 2021; Ammari et al., 2022) aim to predict synthetic contrast-enhanced (CE) standard-dose images from pre-contrast and low-dose CE pairs. However, these approaches tend to blur images unrealistically. Thus, distributional losses (Pasumarthi et al., 2021; Pinetz et al., 2023) are typically used to balance noise synthesis and texture preservation. Unfortunately, these losses tend to hallucinate (Cohen et al., 2018; Antun et al., 2020), which potentially could lead to the generation of false positive CE signals (Haase et al., 2023).

---

[*] Contributed equally

. This short paper is a summary of the preprint by Pinetz et al. (2024).

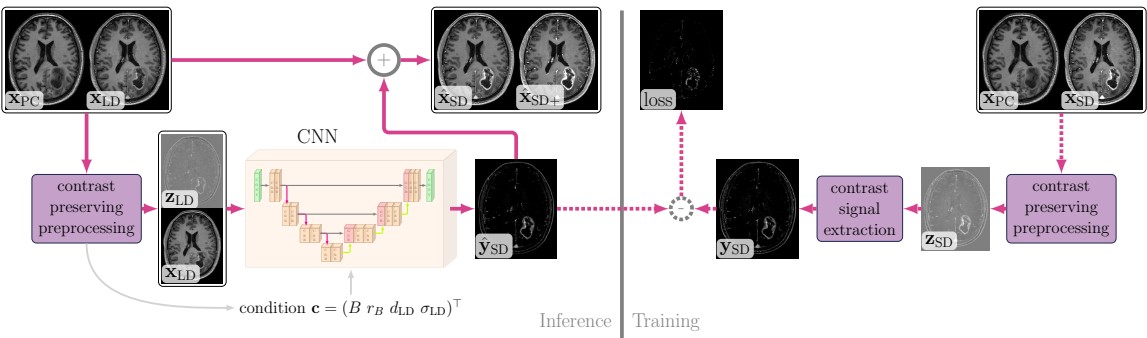

Figure 1: Visualization of our dose reduction and contrast enhancement DL approach.

## 2. Method

In this work, we disentangle GBCA signal enhancement from synthesizing realistic images by focusing on the differential contrast signal encoded in subtraction images of pre- and post-contrast T1-weighted MRI images. In detail, we consider *pre-contrast* $\mathbf{x}_{\mathrm{PC}} \in \mathbb{R}^n$ images and post-contrast images acquired after administering the *standard-dose* $\mathbf{x}_{\mathrm{SD}} \in \mathbb{R}^n$ or a *lower-dose* $\mathbf{x}_{\mathrm{LD}} \in \mathbb{R}^n$. We fix the weight-dependent standard dose to $d_{\mathrm{SD}} = 1$. Hence, low-dose CE images are acquired with dose $d_{\mathrm{LD}} \in (0, 1)$. Figure 1 illustrates the three essential steps of our novel approach.

To extract the differential contrast signal, we first register the pre-contrast $\mathbf{x}_{\mathrm{PC}}$ and post-contrast $\mathbf{x}_{\{\mathrm{LD,SD}\}}$ images rigidly and radiometrically as performed by Pinetz et al. (2023). To harmonize the noise of the initial subtraction images $\tilde{\mathbf{z}}_{\mathrm{LD}} = \mathbf{x}_{\mathrm{LD}} - \mathbf{x}_{\mathrm{PC}}$ and $\tilde{\mathbf{z}}_{\mathrm{SD}} = \mathbf{x}_{\mathrm{SD}} - \mathbf{x}_{\mathrm{PC}}$, we roughly estimate the noise levels $\sigma_{\mathrm{LD}}, \sigma_{\mathrm{SD}}$ using negative intensities in the brain since these are not affected by the contrast agent. Then, we robustly estimate the local mean $\mathbf{m}_{\{\mathrm{LD,SD}\}} \in \mathbb{R}^n$ and standard deviation $\mathbf{s}_{\{\mathrm{LD,SD}\}} \in \mathbb{R}^n$ using large Gaussian filter kernels by masking out outliers. After correcting for spatially varying noise, we multiply both subtraction images by $\sigma_{\mathrm{LD}}$ to homogenize the noise, i.e.

$$\mathbf{z}_{\mathrm{LD}} \coloneqq \frac{\tilde{\mathbf{z}}_{\mathrm{LD}} - \mathbf{m}_{\mathrm{LD}}}{\mathbf{s}_{\mathrm{LD}}} \sigma_{\mathrm{LD}} \approx \sigma_{\mathrm{LD}} \frac{\tilde{\mathbf{z}}_{\mathrm{SD}} - \mathbf{m}_{\mathrm{SD}}}{\mathbf{s}_{\mathrm{SD}}} =: \mathbf{z}_{\mathrm{SD}}. \tag{1}$$

As a result, the two subtraction images differ essentially only in the CE regions.

Next, we compute the *contrast signal* image $\mathbf{y}_{\mathrm{SD}}$ from the standard-dose subtraction image $\mathbf{z}_{\mathrm{SD}}$. This image serves as the ground truth for training the CNN. To avoid synthesizing noise, we extract the CE mask $\mathbf{p}_{\mathrm{SD}} = f(\mathbf{z}_{\mathrm{SD}}) \coloneqq \mathrm{sigmoid}(w\mathbf{z}_{\mathrm{SD}} + b) \in [0, 1]^n$. Here $w, b \in \mathbb{R}$ are chosen to trade-off noise suppression and preservation of faint CE signals. Applying this mask to the standard-dose subtraction image, we get the target image $\mathbf{y}_{\mathrm{SD}} = \mathbf{p}_{\mathrm{SD}} \odot \mathbf{z}_{\mathrm{SD}}$.

Using the low-dose subtraction image $\mathbf{z}_{\mathrm{LD}}$ along with $\mathbf{x}_{\mathrm{LD}}$ as inputs, we train a CNN to predict the associated contrast signal $\hat{\mathbf{y}}_{\mathrm{SD}}$. Consequently, the task of the CNN is to suppress noise and artifacts and increase the contrast signal of the low-dose subtraction image $\mathbf{z}_{\mathrm{LD}}$. The CNN is conditioned on metadata to account for physical parameters affecting the CE behavior. Along the administered dose $d_{\mathrm{LD}}$, we consider the scanner's field strength $B \in \{1.5, 3\}$ and the GBCA's relaxivity $r_B \in \mathbb{R}_+$. In addition, we include an estimate of the noise level $\sigma_{\mathrm{LD}}$ of the low-dose subtraction image.

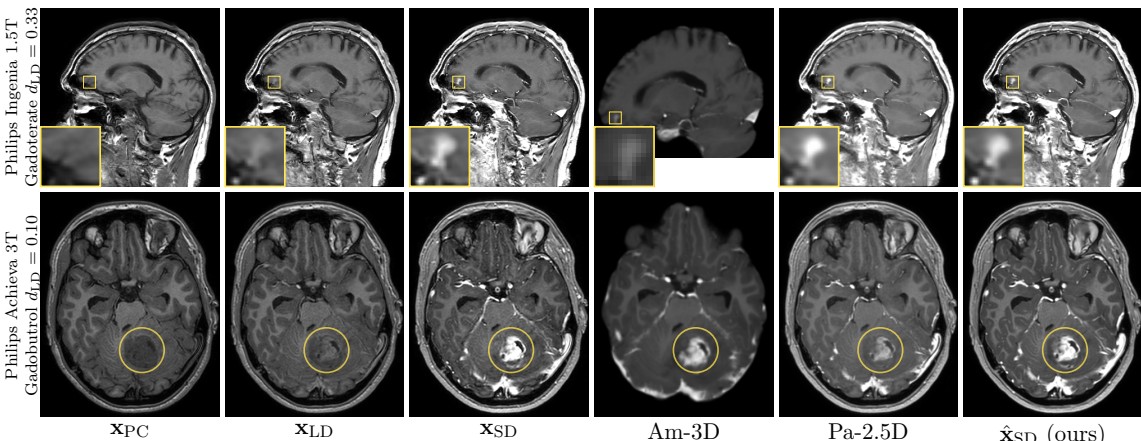

Figure 2: The top row depicts a 33% low-dose sample, where the yellow zoom highlights a
lesion. The bottom row shows a case with a 10% low-dose image and the lesion
is marked by the yellow circles.

To focus the learning problem on CE regions and the brain, we used a registered brain
atlas mask and the CE mask $\mathbf{p}_{SD}$ to weigh the Huber distance. We trained all competing
models on 608 PC, LD, and SD T1-weighted images acquired with Philips Ingenia 1.5/3T
and Achieva 1.5/3T scanners using different GBCA types and dose levels prospectively
collected at the University Hospital Bonn. Further details, regarding the different steps,
training and dataset can be found in the preprint by Pinetz et al. (2024).

## 3. Results

A qualitative comparison of Ammari et al. (2022) (Am-3D), Pasumarthi et al. (2021) (Pa-
2.5D), and our approach on two test samples is shown in Figure 2. The CE signal strength
in pathological regions (highlighted by the yellow circles) is well visible for Am-3D, despite
strong blurring and non-linear intensity transform. Pa-2.5D yields better image quality but
the contrast strength in pathological regions is not well captured (overshooting at the top
and undershooting at the bottom). In contrast, our approach yields the best image quality
and predicts the CE more accurately. For a more detailed numerical evaluation of synthetic
and real low-dose datasets and external data, we refer to the associated preprint (Pinetz
et al., 2024).

## 4. Conclusion

In this abstract, we introduced a novel method for reducing GBCA dosage by leveraging
subtraction images. Our conditional CNN effectively cleans and enhances degraded subtrac-
tion images derived from low-dose and pre-contrast images to extract the contrast signal.
Furthermore, incorporating meta-information from the acquired images, such as the type
and dosage of the injected GBCA or the scanner's field strength is beneficial for predicting
the enhancement strength more accurately.

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
