# OpenReview forum: "Differential contrast enhancement using conditional deep learning for Gadolinium dose reduction in brain MRI"
_MIDL.io/2024/Short_Papers — MIDL 2024 Short Papers_

### Official Review · Reviewer_aXXK · 2024-04-24

**Confidence:** 5
**Final Rating:** 4

**Review:**

pre-contrast / post confusing
noise
relevance

In this well-written paper, the authors present a method to estimate a contrast-enhanced (CE) brain MRI from a pre-contrast image and a low-dose scan. This is an application with high clinical relevance, in the quest of minimizing dosage in CE-MRI. The method itself is a Unet that predicts a residual; the novelty stems from the careful "contrast-preserving" preprocessing used go generate the training data, with enables extracting  accurate residuals from image pairs. Qualitative results are shown in Figure 2, whereas quantitative results are provided in a separate preprint.

Two minor comments:

PC for "pre-contrast" (vs post-contrast) is a bit confusing. Maybe use a different nomenclature / abbreviation?

Is it really that bad to try to predict noise? The network will be unable to do it and simply ignore it, no?

---

### Decision · Program_Chairs · 2024-04-26

Accept